# Effects of One-Step Abrupt Temperature Change on Anaerobic Co-Digestion of Kitchen Waste with Dewatered Sludge

**Weijie Hu [1], Youfei Zhou [1], Hong Zhu [2] and Tianfeng Wang [2,*]**

1   Design Institute No.3, Shanghai Municipal Engineering Design and Research Institute (Group) Co., Ltd., Shanghai 200092, China; huweijie@smedi.com (W.H.); feiyouzhou@163.com (Y.Z.)
2   College of Petrochemical Engineering, Lanzhou University of Technology, Lanzhou 730050, China; skylarpooot@163.com
*   Correspondence: wangtianfeng@lut.edu.cn or gloomysmile@163.com

**Abstract:** The operating temperature of anaerobic digesters should be adjusted to adapt to seasonal variations in environmental temperature and the composition of organic solid waste. This study investigated the effects of one-step abrupt temperature changes (from mesophilic to thermophilic temperature, M–T, and from thermophilic to mesophilic temperature, T–M) and the inoculation ratio on methane yield and microbial diversity during the anaerobic co-digestion of kitchen waste with dewatered sludge. The results showed that the cumulative methane yield (CMY) level resulting from thermophilic control and the M–T digesters was greater than that resulting from mesophilic control and the T–M digesters. The CMF of M–T digesters increased, whereas the CMY of T–M digesters gradually decreased with an increase in the inoculation ratio. The maximal CMY was 385.1 mL/g-VSS$_{added}$, which corresponded to an M–T digester with a 5% inoculation ratio. In the later stage of anaerobic digestion, the bacterial community of T–M was more diverse than that of M–T, but the archaeal community of M–T was more diverse than that of T–M. The one-step temperature change from thermophilic to mesophilic temperature was more stable than that from mesophilic to thermophilic temperature.

**Keywords:** ammonia; inoculum ration; methane yield; bacteria; archaea

## 1. Introduction

The world annually produces approximately 1.3 billion tons of kitchen waste [1]. KW is prone to rot and contains various pathogenic microorganisms and parasites that can lead to the spread of diseases [2]. KW is characterized by its high moisture content, organic matter content, and carbon-to-nitrogen ratio (C/N ratio) [3,4]. Consequently, KW is well-suited for anaerobic digestion (AD) treatment [2,3,5]. Under thermophilic conditions, semi-continuous, dry-mode anaerobic digestion systems for KW will achieve a faster steady-state operation and will remain stable [6]. However, improper operation (such as fluctuations in temperature and substrate) may result in digester instability [7]. The inhibition of organic acids or an imbalance in microbial communities can result in decreased performance and degradation efficiency of the digesters [8].

China annually produces approximately 39 million tons of excess sludge, with a moisture content of 80% (dewatered sludge, DS), from wastewater treatment plants [9]. Due to the accumulation of harmful substances in DS, like resistance genes, pathogens, and heavy metals, proper disposal measures are necessary to minimize harm to the environment [10,11]. AD is a crucial technology for the sustainable recovery of energy and the prevention of environmental pollution caused by sludge disposal. AD not only stabilizes DS but also generates methane-rich biogas (with a methane content of approximately 65%) [12,13]. However, the slow hydrolysis process and low C/N ratio associated with sludge AD will result in a much lower methane yield than the theoretical methane yield [14].

The relatively low C/N ratio in DS makes it easy for ammonia inhibition to happen, whereas the higher C/N ratio in KW complements that of DS and reduces the possibility of ammonia inhibition [15]. Meanwhile, KW and DS complement each other in terms of their characteristics, including their water content and microbial community structure [16]. The combination of KW and DS provides improved stability for co-digestion systems [17,18]. Temperature is an important factor that influences methane production in AD [19]. Increasing the temperature during anaerobic digestion enhances microbial activity and growth, leading to the rapid breakdown of organic matter into smaller molecules and their further stabilization into valuable organic fertilizers and biogas [12]. It also promotes higher digestion rates and the growth of favorable microbial flora, resulting in more complete organic matter digestion [20]. Dramatic temperature changes or frequent fluctuations can affect bacteria, especially methanogenic bacteria. Temperature fluctuations of more than 1.1 °C per day can lead to AD failure [12].

Usually, anaerobic digestion is carried out at a constant temperature [21]. However, the operating temperature of anaerobic digesters over time should balance energy consumption and system stability [22]. The operating temperature of anaerobic digesters should be adjusted to adapt to seasonal variations in environmental temperature and the composition of organic solid waste [23].

In conventional sludge AD, with the transition from mesophilic to thermophilic temperature, the thermophilic microbial community was established from a mesophilic digester [24]. The thermophilic anaerobic digester remained stable for 20 days after a one-step temperature increase from a mesophilic digester. After a one-step temperature increase, there was a rapid proliferation of *Methanosarcina*, *Methanothermobacter*, and *Methanoculleus* within 11 days. However, the sludge was rich in microorganisms with low concentrations of ammonia, which is beneficial for the normal metabolism of methanogens [25]. During the thermophilic anaerobic digestion of garbage slurry inoculated with mesophilic-waste-activated sludge, thermophilic methanogens could be enriched [26]. There was a similar microbial community in the enriched microbiome at different concentrations of substrate. *Methanothermobacter*, *Methanosarcina*, and other thermophilic bacteria were enriched in the community over time. However, there is limited research on the effects of one-step abrupt temperature changes (from mesophilic to thermophilic temperature, M–T, and from thermophilic to mesophilic temperature, T–M) on the anaerobic co-digestion system of KW and DS [27]. This information is of significant use for the application of temperature changes in the anaerobic digestion of sludge across different seasons in areas experiencing large temperature variations between winter and summer.

This study implemented the anaerobic co-digestion of KW and DS. The mixture used was inoculated with sludge acclimated to both mesophilic and thermophilic conditions in the event of an abrupt temperature change. The objectives of this study were as follows: (1) compare anaerobic digestion performance under mesophilic and thermophilic temperatures; (2) investigate the effects of an abrupt temperature change on methane yield and the microbial community in the anaerobic co-digestion system.

## 2. Materials and Methods

### 2.1. Materials and Pretreatment

The KW used in the experiment was sourced from Gansu Chinai Bioenergy System Co., Ltd, Lanzhou, China. Before the experiment, the KW underwent pretreatment, including the removal of plastic, bones, paper towels, and chopsticks. It was then crushed, stirred, and grounded. To prevent the sampling tube from becoming clogged, the KW was passed through a 0.4 mm iron screen. After pretreatment, the KW was stored in a plastic bottle in the refrigerator at −20 °C for subsequent use in the experiments.

The excess sludge and DS used in this experiment were sampled from a sewage treatment plant in Lanzhou, Gansu, which with a sewage capacity of 100,000 m³/day and the utilization of an AAO + MBR (Anaerobic-Anoxic-Oxic + Membrane Bio-digester) process excess sludge was collected from the secondary sedimentation tank, whereas DS was col-

lected from the centrifugal dehydrator. The excess sludge was acclimated in $35 \pm 1\ °C$ and $55 \pm 1\ °C$ thermostat water baths for 28 days. This process yielded mesophilic acclimated sludge (MAS) and thermophilic acclimated sludge (TAS) for subsequent inoculation.

The basic properties of KW and sludge are shown in Table 1. Both DS and KW have high TS, but KW has higher VS compared with DS. Hence, KW has a high level of SCOD pH below the threshold for methanogenic bacteria activity.

**Table 1.** Basic properties of KW and sludge.

| | MAS | TAS | DS | KW |
|---|---|---|---|---|
| TS (%) | $1.67 \pm 0.01$ | $1.48 \pm 0.03$ | $24.75 \pm 0.13$ | $23.76 \pm 0.31$ |
| VS (%) | $0.92 \pm 0.01$ | $0.80 \pm 0.02$ | $10.23 \pm 0.11$ | $18.99 \pm 0.27$ |
| pH | $7.48 \pm 0.21$ | $7.69 \pm 0.15$ | | $5.20 \pm 1.06$ |
| SCOD (mg/L) | $2926.2 \pm 22.4$ | $3850.7 \pm 62.0$ | | $80{,}644.8 \pm 903.7$ |
| TA (mg/L) | $1860.2 \pm 0.0$ | $2340.2 \pm 22.2$ | | $594.4 \pm 60.4$ |
| TAN (mg/L) | $493.6 \pm 4.6$ | $385.62 \pm 62.0$ | | $423.5 \pm 0.5$ |

Note: Mesophilic acclimated sludge, MAS. Thermophilic acclimated sludge, TAS. Total solids, TS. Volatile solids, VS. Soluble chemical oxygen demand, SCOD. Total alkalinity, TA. Total ammonia nitrogen concentration, TAN.

## 2.2. Experimental Design

A total of 300 g KW was mixed with 150 g DS as a substrate for digestion, and 300 g MAS or TAS was added as inoculum. M0 and T0 were cultured in a thermostatic water bath at $35 \pm 1\ °C$ or $55 \pm 1\ °C$ until the end of the experiment. T1–T5 and M1–M5 were cultured in a thermostatic water bath at $35 \pm 1\ °C$ or $55 \pm 1\ °C$, respectively. The abrupt temperature experiment started on the tenth day, with a raised $35 \pm 1\ °C$ to $55 \pm 1\ °C$ and a lowered $55 \pm 1\ °C$ to $35 \pm 1\ °C$. Immediately afterward, 0%, 5%, 10%, 15%, and 20% TAS or MAS were added to the T1–T5 and M1–M5 experimental groups, respectively (Table 2). During the initial 10 days, 4 M NaOH was used to adjust the pH to 7.0–7.5.

**Table 2.** Experimental setup.

| Temperature Change | Ratio of Inoculation after Temperature Change | | | | |
|---|---|---|---|---|---|
| | 0% | 5% | 10% | 15% | 20% |
| Always 35 °C | M0 | | | | |
| Always 55 °C | T0 | | | | |
| Change from 35 °C to 55 °C (Inoculation with TAS on day 10) | T1 | T2 | T3 | T4 | T5 |
| Change from 55 °C to 35 °C (Inoculation with MAS on day 10) | M1 | M2 | M3 | M4 | M5 |

The digesters were glass bottles sealed by rubber septa with a 2 L working volume and two small holes through the lid (with a 6 mm inner diameter), which were connected to the bag and the peristaltic pump, respectively, through soft silicone tubes (with a 6 mm inner diameter) for collecting gas or digestate samples. To ensure that the co-substrate in each digester was mixed evenly, each digester needed to be shaken for 1–2 min every morning. All processes were repeated three times to ensure accuracy.

## 2.3. Analytical Methods

The pH, TS, VS, SCOD, TAN, and TA determination methods were derived from the Standard methods [28]. The concentration of free ammonia nitrogen (FAN) in the sample was determined in accordance with the formula provided by Hansen et al. [29]. The volume of biogas was measured by the wet anticorrosive gas flowmeter (LMF-2, Alpha, Shanghai, China), and the biogas composition was measured by the biogas analyzer (Biogas5000, Geotach, Coventry, UK). The modified Gompertz equation was used to describe the methane yield in the experiment [30]. Volatile fatty acid (VFA) was measured by a gas chromatograph (SP-7890 Plus, Ruihong, Tengzhou, China).

To elucidate the microbial communities, high-throughput sequencing technology was employed. Different letters were used to represent the microbial community analysis results at three different times (a: on the 9th day, b: on the 27th day, c: on the 43rd day). Microbial DNA was extracted from the solid phase of anaerobic digestion samples using the Illumina MiSeq platform (Illumina, San Diego, CA, USA) and the soil DNA kit (E.Z.N.A.® Soil DNA kit, Omega Bio-tek, Norcross, GA, USA). The V3-V4 hypervariable region of the bacterial 16S rRNA gene was amplified using primers 806R (5′-GGACTACHVGGGTWTCTAAT-3′) and 338F (5′-ACTCCTACGGGAGGCAGCAG-3′). Species information classification and integration of the original data were completed with reference to relevant websites, and afterward, the OTU (operational taxonomic unit) table was generated. For detailed experimental procedures and data preprocessing methods, please refer to the previous study [31].

*2.4. Statistical Analysis*

SPSS for Windows version 22.0 was used to analyze the correlation between environmental factors and microbial communities (*, $p < 0.05$; **, $p < 0.01$; ***, $p < 0.001$). Origin for Windows version 2022 was used to create all figures.

## 3. Results and Discussion

*3.1. Digestion Performance*

### 3.1.1. pH, TAN, FAN

The pH value was an important parameter to evaluate the stability of digesters [32]. The optimal pH range for anaerobic digestion was 6.5 to 8.0 [33]. The overall pH value changing trends of T1–T5 experimental groups were relatively gentle after inoculation with TAS (Figure 1a). The T1-T5 pH values were between 7.0 and 8.0, which is a relatively stable level and within the optimal range of active metabolism of AD [26]. The pH change in M1–M5 showed a decreasing trend at first and then continued to increase (Figure 1b). This phenomenon might be due to the rapid increase of microbial activity in the early stage of digestion, which consumed organic matter and produced a large number of acidic metabolites [20]. In addition, some microorganisms were more active at lower temperatures and produced more alkaline metabolites, which caused an increase in pH [34]. In the early stage of digestion, the pH values of M0 and T0 experienced a sharp decrease. This can be attributed to the limited activity of methanogens during the hydrolysis and acidification stage [35]. As a result, VFA was not readily utilized by methanogens to produce methane, leading to the accumulation of VFA in the digestion system [8]. The pH value was adjusted by adding 4 M NaOH before the inoculation experiment so that it was kept within the range of 6.5–7.5. With the continuous proliferation of methanogens, VFA will be consumed continuously, which ultimately leads to the constant rise of the pH of T0 [8].

In general, in a sludge/food waste anaerobic co-digestion system, a higher proportion of sludge in the digester resulted in a higher pH [36]. This may be due to the increased proportion of sludge, which led to an increased buffering capacity of the system, so the pH-changing trend did not show an obvious deviation [37]. However, the pH of M1–M5 did not change with the increase of the TAS inoculum ratio. The pH under mesophilic conditions (M0) rose more slowly than that under thermophilic conditions (T0). Thermophilic conditions promoted the release of ammonia nitrogen while accelerating the consumption of organic acids by microorganisms, both of which were responsible for the rapid increase in pH under thermophilic conditions [38]. After adjusting pH, the overall pH-changing trend in the experimental and control groups was similar to that in previous studies [14,39]. However, the minimum pH value (7.1) in this study was higher than that in this experiment (6.9) under thermophilic conditions, which might be due to the higher VS/TS of the substrate in this experiment, allowing the formation of organic acids leads to a lower pH [37].

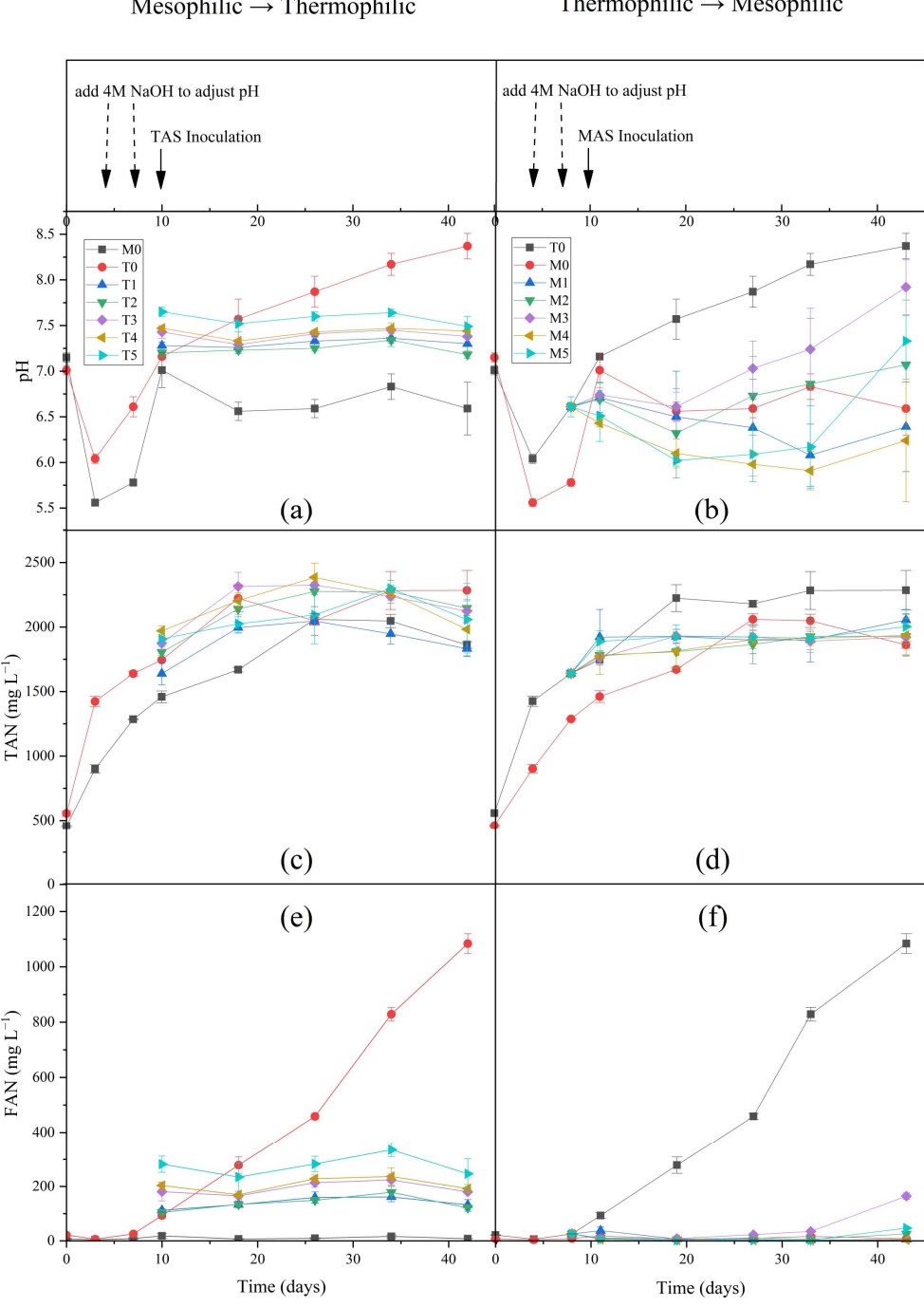

**Figure 1.** Changes in pH (**a**,**b**), TAN (**c**,**d**), and FAN (**e**,**f**) during the co-digestion process.

The TAN level in both the experimental and control groups was between 1500 and 2500 mg $L^{-1}$ at the end of digestion (Figure 1c,d), indicating that all methanogens in this experiment might have been affected by ammonia inhibition. The TAN changing trend of all digesters showed a rapid increase in the early stage of digestion, which was consistent with previous research results [32,40]. KW contained complex macromolecules with slow degradation rates, such as proteins and carbohydrates, which could lead to a high concentration of TAN during the digestion process [41]. The high concentration of TAN could inhibit the activity of methanogens, leading to the accumulation of VFA, a decrease in pH, and lower methane production. However, a low concentration of ammonia could provide sufficient buffering capacity and essential nutrients for microbial growth [19]. When the TAN concentration exceeded 1500 mg $L^{-1}$, ammonia could inhibit the growth

of methanogens, and when the TAN concentration exceeded 2500 mg $L^{-1}$, methanogens could be significantly inhibited [42]. In the early stages of digestion, the breakdown of organic matter and the metabolism of anaerobic bacteria could produce a significant amount of ammonia nitrogen, but the microbial community in the anaerobic digester was still in the adaptation and adjustment phase, and could not fully utilize the generated ammonia nitrogen [43]. As digestion progressed, the anaerobic bacteria gradually adapted to the ammonia nitrogen and used it for growth and reproduction [19]. The TAN in the experimental group increased with the sludge inoculum ratio, which was likely due to the accelerative effect of microorganisms in the TAS on the decomposition of KW, leading to more production of TAN.

As digestion progressed, the difference between FAN and TAN became more apparent. FAN was present in TAN, which increased as the digestate pH and digestion temperature increased [29]. Compared with TAN, FAN had direct toxicity to methanogens and could penetrate microbial cell membranes, causing damage to enzyme systems [44]. By comparing the changes in TAN and FAN levels in the control group (Figure 1e,f), it was found that the TAN and FAN levels in the thermophilic digesters were significantly higher than those in the mesophilic digesters, and with a wider variation range. This was because the thermophilic conditions promoted the degradation of nitrogen-rich substances and the solubility of organic matter in the co-substrate, resulting in more thorough organic matter degradation and a faster rate of TAN production [45]. The TAN level in the experimental group did not show a significant difference compared to the control group, but the FAN level in the experimental group was much lower than that in T0 of the control group. This was mainly because the rise to a higher operating temperature could lead to an increase in ammonia nitrogen, which could inhibit the life activities of anaerobic microorganisms [46]. Moreover, ammonia nitrogen could inhibit the activity of propionic acid-degrading microorganisms, leading to the accumulation of propionic acid [14,39]. Propionic acid, as a difficult-to-degrade VFA, had the most obvious inhibitory effect on methanogens, therefore inhibiting the progress of anaerobic digestion [46].

### 3.1.2. SCOD, VFA, TA, VFA/TA

The SCOD concentration (Figure 2) in the digestate was a measure of the dissolved organic matter in the anaerobic digestion [31]. In general, the SCOD of the experimental groups T1–T5 and M1–M5 exhibited a slow declining trend at different temperatures. As the TAS inoculum ratio of T1-T5 increased, the SCOD gradually decreased, which should be the dilution effect of the low SCOD (VSS) of TAS. As the MAS inoculum ratio of M1-M5 increased, the SCOD only gradually decreased in the initial period after inoculation. Different methane yields resulted in different SCOD levels.

The sudden increase in SCOD in the early stage in M0 and T0 was mainly attributed to the hydrolysis of macromolecular organic matter in kitchen waste and DS into smaller molecules by bacterial extracellular enzymes [40]. Additionally, the addition of NaOH might have further promoted bacterial hydrolysis, resulting in a sudden increase in SCOD [36]. The decline trend of SCOD in M1–M5 was faster than that in T1–T5, which should be related to the high methane yield of M1–M5. The decrease in SCOD indicated the utilization of organic matter by methanogens and the steady functioning of the digestion system. The decrease in SCOD was faster in T0 compared with M0, suggesting that thermophilic temperatures could expedite the decomposition of organic matter. As to T1–T5, the SCOD was at a high level, which should be related to the low methane yield. Low methane yield was caused by the synergistic inhibition of high FAN (Figure 1) and high VFA (Figure 3). The synergistic effect of FAN and VFA could cause severe inhibition of methanogens [35].

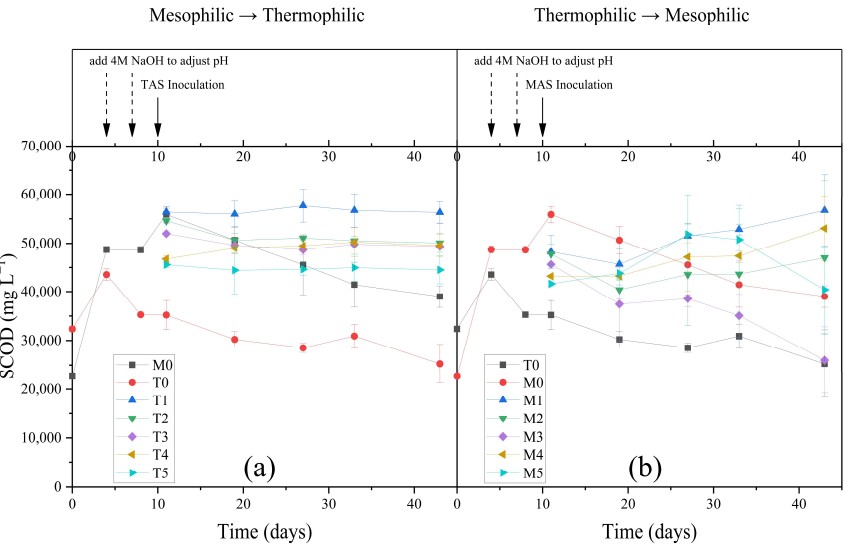

**Figure 2.** Changes in SCOD ((**a**), Mesophilic to Thermophilic; (**b**), Thermophilic to Mesophilic) during the co-digestion process.

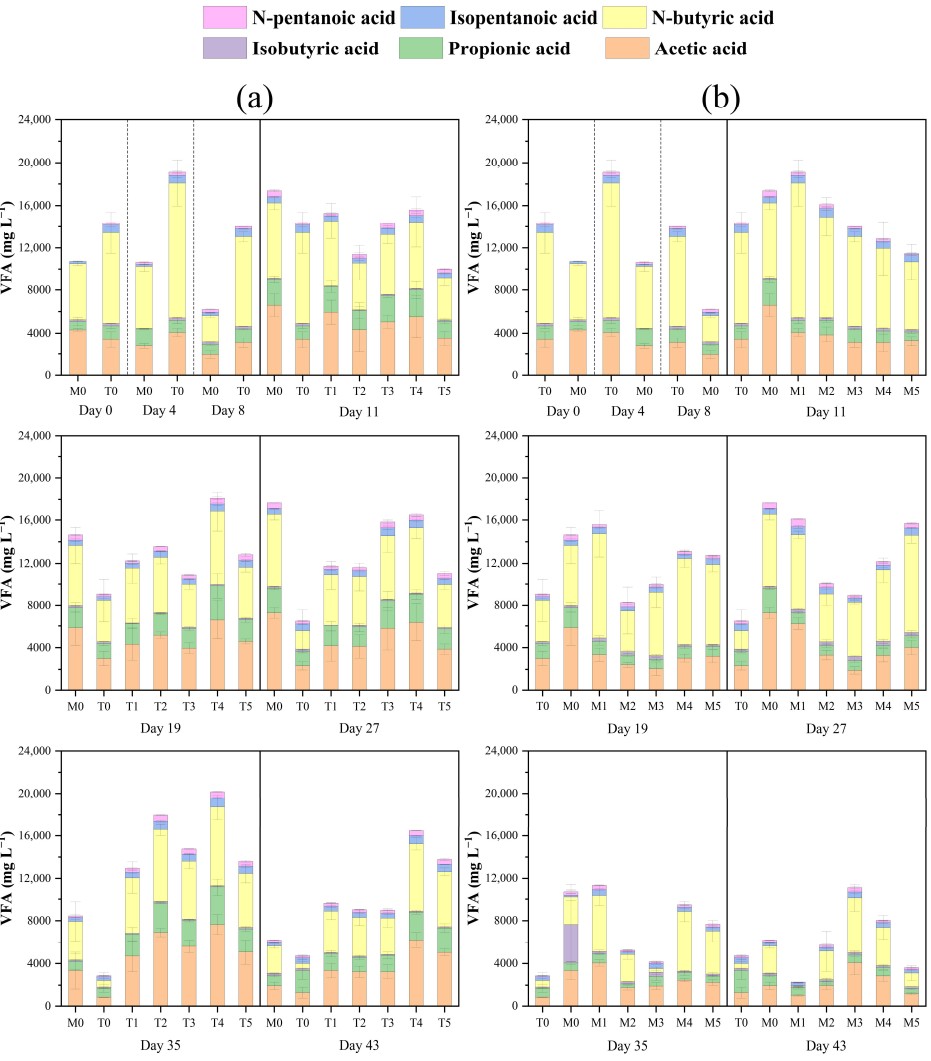

**Figure 3.** Changes in VFA ((**a**), Mesophilic to Thermophilic; (**b**), Thermophilic to Mesophilic) during the co-digestion process.

VFA concentration was an important indicator, which reflected the extent of organic matter hydrolysis and acidification in the anaerobic digestion [47]. The VFA of M0 showed an increasing and then decreasing trend, reaching a maximum of 17,370 mg L$^{-1}$ on the 11th day, which was consistent with the trend of SCOD concentration. In the early stage of anaerobic digestion, proteins, carbohydrates, and long-chain fatty acids in DS and KW were converted to VFA, which accumulated in the anaerobic digesters [47]. The VFA of T0 digesters showed a similar increasing and then decreasing trend as M0, reaching a maximum of 18,388 mg L$^{-1}$ on the eighth day. However, the overall VFA level of T0 was lower than that of M0. This was because at thermophilic temperatures, the anaerobic microbial decomposition and metabolic rate increased, and more metabolic products were converted to methane [19]. Therefore, the relative contribution levels of VFA were relatively low. Additionally, some mesophilic microbial communities also showed higher growth rates and metabolic activity in organic-rich waste such as KW [48].

The VFA of T1–T5 was higher than that of T0. However, despite the high VFA level, the pH of T4 digesters remained within the reasonable range. This suggested that the microbial community in the digesters was able to handle the VFA accumulation effectively, thus avoiding the problem of acidification. In general, the VFA of M1–M5 decreased faster than that of T1–T5, especially on the 35d and 43d. On the 19th day, the VFA of T1, T2, and T3 had decreased compared with the 11th day, primarily because of the slight gas production during the first two days after inoculation with TAS, where microbial activity consumed some of the VFA. However, the VFA of T4 and T5 continued to increase, indicating that VFA was still accumulating and not being utilized by the microbial community [40]. On the 19th, 27th, and 35th day, the VFA remained high, but there was no methane production. This suggested that the digesters were inhibited by the accumulating VFA and FAN [19].

TA could characterize the buffering capacity of the anaerobic digestion system [38,49]. In the first 11 days, the TA of M0 continued to increase (Figure 4), partly due to the artificial addition of NaOH, while the ammonia nitrogen produced during the reaction process offset part of VFA. The TA of T0 had been fluctuating around 7000 mg/L, which was relatively stable. After inoculation with TAS, the change of TA in all digesters was consistent, but there were slight differences. Finally, the TA of T1–T5 was between 10,000 and 11,000 mg/L. TA of M1–M5 was lower than that of T1–T5, which may be due to the fact that the lowering digestion temperature has little effect on the decomposition of organic matter by microorganisms [41].

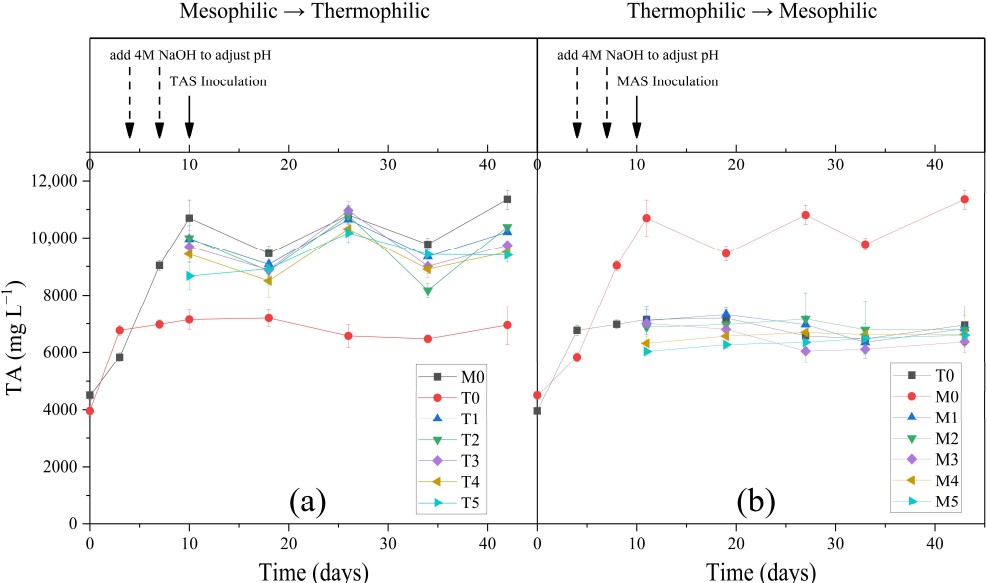

**Figure 4.** Changes in TA ((**a**), Mesophilic to Thermophilic; (**b**), Thermophilic to Mesophilic) during the co-digestion process.

Total volatile fatty acids/total alkalinity (TVFA/TA, Figure 5) was a necessary index to judge the stability of the digester [38]. TVFA/TA was a parameter used to describe acid-base balance during anaerobic digestion [33]. When the TVFA/TA value was less than 0.35, the TA was sufficient to buffer the acidity of TVFA concentration [50], thus maintaining good acid-base balance and relative stability of the digester. However, if the TVFA/TA value exceeded 0.8, indicating that the VFA concentration is high, it might lead to acidification during the digestion process and seriously affect the efficiency and stability of anaerobic digestion [33,38]. TVFA/TA of T1–T5 was significantly greater than that of M1–M5. Moreover, TVFA/TA of T1–T5 was greater than 0.8 almost throughout the co-digestion process. However, TVFA/TA of M1–M5 gradually decreased as the co-digestion progressed, which was less than 0.8 (even 0.35) in the later stage (except M4). These results indicated that M1–M5 was a normal and stable co-digestion process.

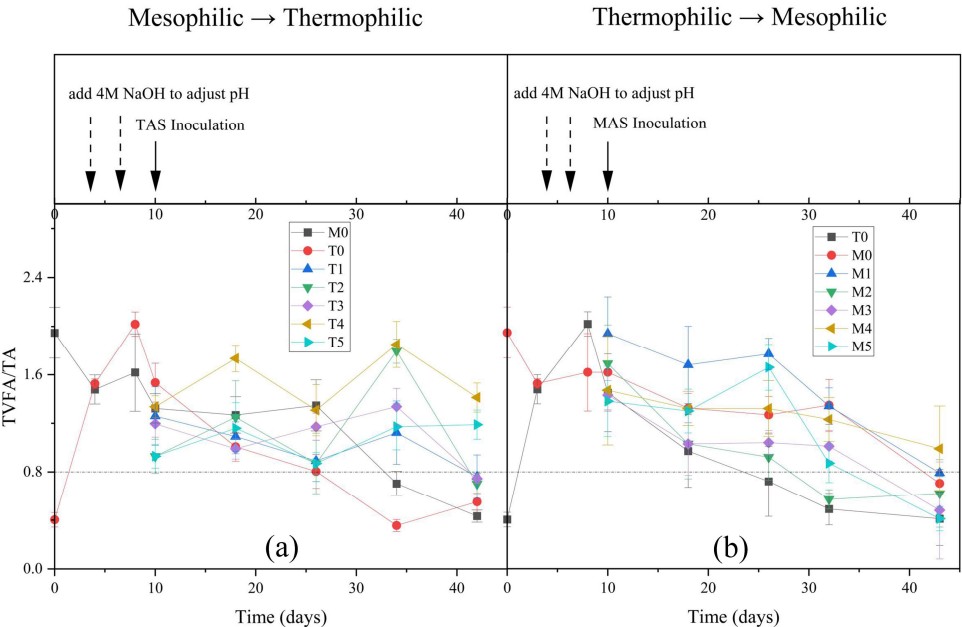

**Figure 5.** Changes in TVFA/TA ((**a**), Mesophilic to Thermophilic; (**b**), Thermophilic to Mesophilic) during the co-digestion process.

### 3.2. Methane Yield

As to T0, the cumulative methane yield (CMY) was 352.0 mL g-VSS$_{added}$ (Figure 6a). As to M0, the CMY was only 32.4 mL g-VSS$_{added}$. As to T1–T5, methane was produced in the first two days after inoculation and after 40 days. The CMY of T1–T5 increased with the increase of the TAS inoculation ratio. This phenomenon should be related to the synergistic inhibition of high FAN and high VFA. The CMY was directly related to the number of methanogens [19,49]. The CMY of T0 was much higher than that of M0. This was because, at the same organic load, the thermophilic temperatures could promote the growth and reproduction of methanogens in the anaerobic digester and improve their activity and metabolic rate within a certain temperature range [19,40]. In addition, the thermophilic anaerobic digesters contained some rare, unique microbial populations that could efficiently utilize some difficult-to-biodegrade organic matters from KW under thermophilic conditions and produce a large amount of methane [49].

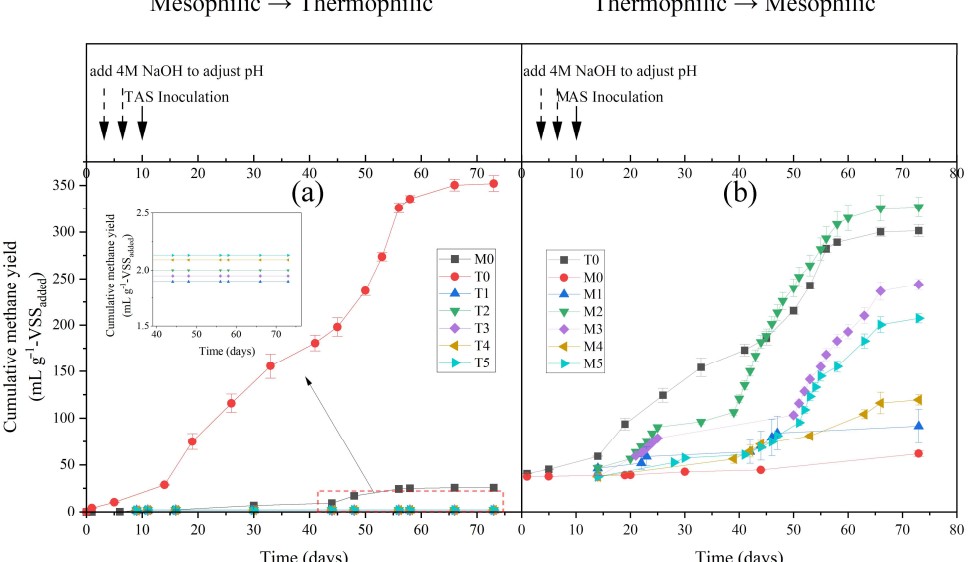

**Figure 6.** Change in cumulative methane yield ((**a**), Mesophilic to Thermophilic; (**b**), Thermophilic to Mesophilic) after inoculation with TAS.

The CMY of M1–M5 were 72.2, 385.1, 275.4, 109.1, and 226.4 mL g$^{-1}$-VSS$_{added}$, respectively. The CMY of M2 with a MAS inoculation ratio of 5% was higher than that of all other digesters. In addition, when the MAS inoculation ratio was higher than 5%, the CMY gradually decreased with the increase of the MAS inoculation ratio. This phenomenon should be related to the special properties of MAS. MAS contained both methanogens and inhibitors [51]. The higher inoculation ratios might have brought about higher concentrations of inhibitors, whereas the low inoculation ratio of inhibitors was diluted by the substrate and introduced methanogenic bacteria to the substrate [19,40].

### 3.3. Microbial Community

#### 3.3.1. Bacterial Community

On the ninth day, at mesophilic temperature (M0-a), the bacterial community was diverse (Figure 7a). The RA of bacteria above 5% were *Bacillus* (6.1%), *Coprothermobacter* (13.7%), *NK4A214_group* (6.0%), and *Sporanaerobacter_acetigenes* (7.5%). However, at thermophilic temperature (T0-a), the RA of bacteria above 5% were only *Coprothermobacter* (33.3%) and *Defluviitoga* (17.0%). *Bacillus* were acid-producing bacteria during anaerobic digestion [52]. *Coprothermobacter* were mainly involved in the acid-producing phase of anaerobic digestion [53]. *Coprothermobacter*, with strong activity at thermophilic temperatures, could cooperate with hydrotropic methanogenic bacteria to degrade organic substrate [54,55]. On the 27th day, as T1–T5, the RA of bacteria above 5% were *Bacillus* (18.2–47.0%), *Coprothermobacter* (6.8–11.1%), and *Acetomicrobium* (1.9–5.1%). With the increased inoculation ratio, the RA of Bacillus first decreased and then increased, but the RA of *Coprothermobacter* and *Acetomicrobium* first increased and then decreased. *Acetomicrobium* could hydrolyze starch, casein, and tributyrin, which could grow at high NH$_3$ levels [53]. The high RA of *Acetomicrobium* in T2–T4 corresponded to the high TAN in T2–T4 (Figure 1c). On the 43rd day, in T1–T5, the bacterial community was diverse. The RA of bacteria above 5% were *Coprothermobacter* (6.7–8.4%) and *Acetomicrobium* (4.6–9.2%). *Syntrophomonas* were syntrophic bacteria [54,55] which had a significant negative correlation with acetic acid, butyric acid, propionic acid, and VFAs (Figure 7b). This phenomenon should be related to the VFA accumulation (Figures 3 and 6a). Enough organic acids slowed down the growth of the related microbes [54,55].

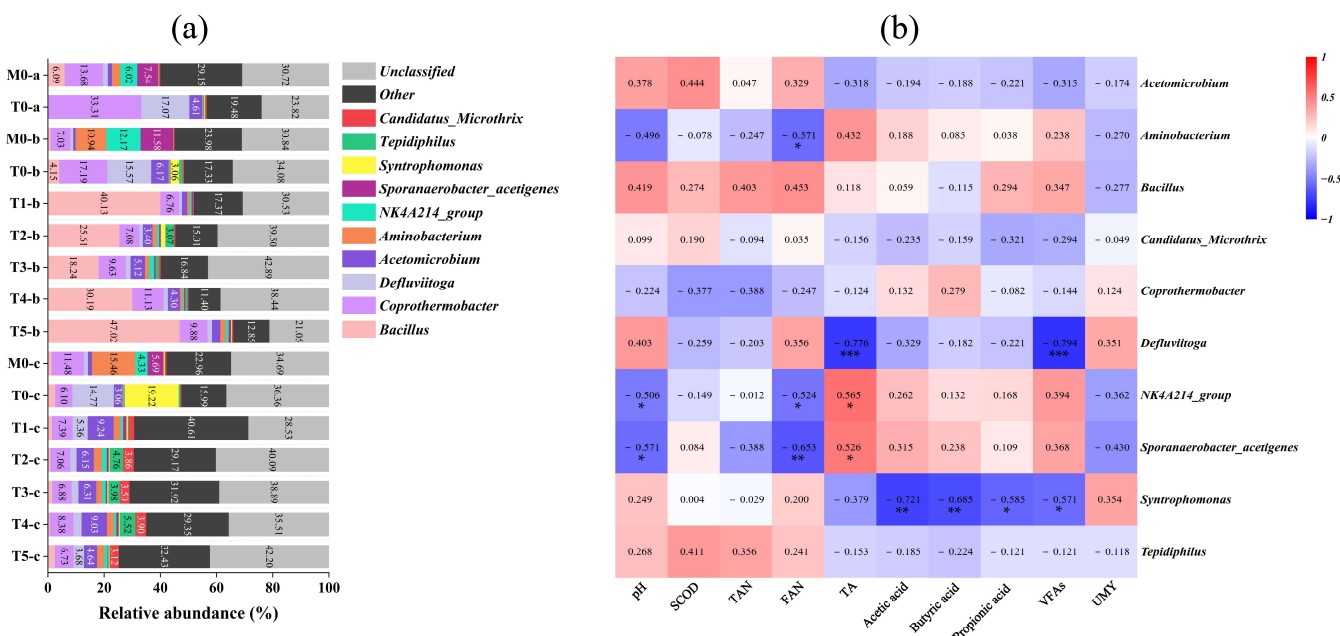

**Figure 7.** Bacterial genus structure (**a**) and correlation heatmap (**b**) of one-step abrupt temperature change from mesophilic to thermophilic. (a: on the 9th day, b: on the 27th day, c: on the 43rd day). *: $p < 0.05$, **: $p < 0.01$; *** $p < 0.001$.

On the 27th day, in M1–M5 (Figure 8a), the RA of bacteria above 5% were *Coprothermobacter* (12.1–32.7%), *Defluviitoga* (15.2–17.8%), *Acetomicrobium* (3.5–6.3%) and *NK4A214_group* (2.2–6.9%). With the increased inoculation ratio, the RA of *Coprothermobacter* decreased. This phenomenon should be related to the RA *Coprothermobacter* of T0 being higher than that of M0. A high inoculation ratio meant a better dilution effect. On the 43rd day, as M1 to M5, the RA of bacteria above 5% were *Coprothermobacter* (8.4–16.2%), *Defluviitoga* (9.5–23.2%), *Acetomicrobium* (2.4–8.9%) and *Aminobacterium* (0.4–6.1%). *Aminobacterium* could degrade many of the amino acids during protein AD [56]. High RA of *Aminobacterium* should be related to the low TAN level of M1–M5 (Figure 1d). In the later stage of AD, the bacterial community of M1 to M5 was more diverse than that of T1–T5. This phenomenon should be related to high methane yield and low VFA of M1 to M5. *Acetomicrobium* were thermophilic acetogenic bacteria [54,55], which had a significant positive correlation with UMY (Figure 8b). This phenomenon, acetic acid, is a key substrate for methane production [54,55]. *Sporanaerobacter acetigenes* were novel acetogenic, facultatively sulfur-reducing bacteria [54,55], which had a significant negative correlation with UMY (Figure 8b). This phenomenon should be related to the inhibition of methanogen activity by hydrogen sulfide [54,55].

### 3.3.2. Archaea Community

On the ninth day, at mesophilic temperature (M0) (Figure 9a), the RA of archaea above 5% were *Methanosaeta* (20.1%), *Methanobacterium* (10.8%), *Methanofollis* (37.5%) and *Methanosarcina* (15.0%). Meanwhile, at thermophilic temperature (T0), the RA of archaea above 5% were *Methanothermobacter* (48.5%), *Methanoculleus* (15.9%), *Methanosaeta* (14.7%) and *Methanospirillum* (5.0%). *Methanosaeta* was highly adaptable to anaerobic environments, which could utilize a wide range of organic substances as substrates for growth and methanogenesis [57]. *Methanobacterium* was a hydrogenotrophic methanogen that could interact with other methanogens and fermenters to promote the conversion of organic matter [58]. *Methanofollis* was a hydrogenotrophic methanogen with an optimum growth temperature of 37 °C [25]. *Methanosarcina* was an acetic acid-nutrient methanogen that typically grew at mesophilic temperatures [57]. *Methanothermobacter* was a hydrogenotrophic methanogen at thermophilic temperatures [59]. *Methanoculleus* was

a hydrogenotrophic methanogen that used $H_2$ as the electron donor to convert $CO_2$ to $CH_4$ [26]. On the 27th day, in T1–T5, the RA of archaea above 5% were *Methanothermobacter* (4.1–28.5%), *Methanoculleus* (3.2–31.0%), *Methanosaeta* (12.9–34.2%), *Methanobacterium* (9.4–35.0%), *Methanofollis* (1.1–12.3%), and *Methanospirillum* (3.0–8.9%). With the increased inoculation ratio, the RA of Methanothermobacter increased. This phenomenon should be related to the high RA of Methanothermobacter in thermophilic inoculation. On 43rd day, in T1-T5, the RA of archaea above 5% were *Methanothermobacter* (4.1–28.5%), *Methanoculleus* (3.2–31.0%), *Methanosaeta* (12.9–34.2%), *Methanobacterium* (1.1–12.3%), *Methanospirillum* (3.0–8.9%), and *Methanobrevibacter* (9.4–35.0%). With the increased inoculation ratio, the RA of *Methanothermobacter* still increased. However, the RA level on the 43rd day was higher than that on the 27th day. The RA of *Candidatus Methanofastidiosum*, *Candidatus Nitrocosmicus*, *Methanobacterium*, *Methanobrevibacter*, *Methanosaeta*, and *Methanospirillum* had a significant negative correlation with UMY (Figure 9b). Only the RA of *Methanothermobacter* had a significant positive correlation with UMY (Figure 9b). Long thermophilic digestion time and low UMY were probably the main reasons for above-mentioned phenomena [54,55].

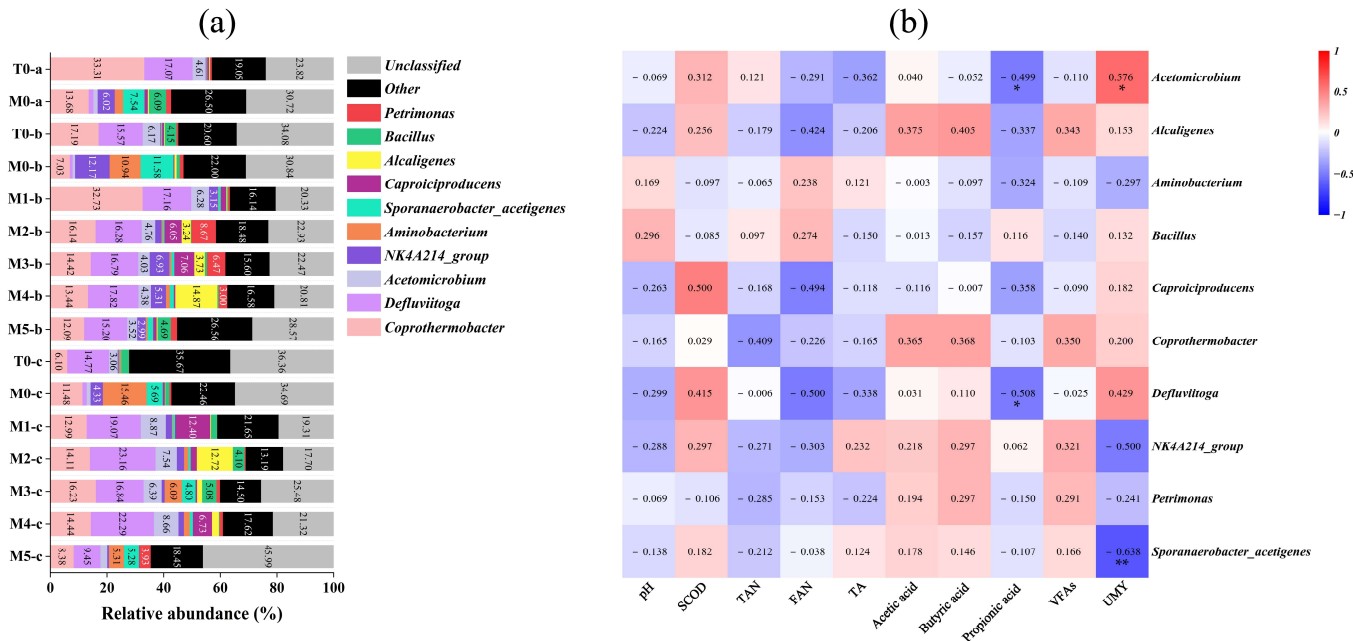

**Figure 8.** Bacterial genus structure (**a**) and correlation heatmap (**b**) of one-step abrupt temperature change from thermophilic to mesophilic. (a: on the 9th day, b: on the 27th day, c: on the 43rd day). *: $p < 0.05$, **: $p < 0.01$.

On the 27th day, in M1–M5, the RA of archaea above 5% were *Methanothermobacter* (26.4–38.3%), *Methanosarcina* (6.2–33.0%), *Methanoculleus* (19.7–44.8%), *Methanosaeta* (0.7–12.7%), *Methanobacterium* (0.8–10.5%), and *Methanospirillum* (0.6–9.9%). With the increased inoculation ratio, the RA of *Methanoculleus* decreased. On the 43rd day, in M1–M5, the RA of archaea above 5% were *Methanothermobacter* (4.2–81.0%), *Methanosarcina* (5.8–81.1%), *Methanoculleus* (1.1–14.4%), and *Methanobacterium* (0.5–34.1%). On the 43rd day, the archaea diversity of T1–T5 was greater than that of M1–M5. This phenomenon should be related to different VFA and methane yields. High VFA levels tended to have more diverse methanogens [60]. Unlike the correlation heatmap of a one-step abrupt temperature change from mesophilic to thermophilic (Figure 9b), only *Methanobrevibacter*, *Methanofollis*, and *Methanosaeta* had a significant negative correlation with UMY (Figure 10b). In fact, the UMY level of M1–M5 was higher than that of T1–T5. The higher UMY of M1–M5 should be implemented with the cooperation of different methanogens [19,40]. Therefore, no methanogen showed a significant positive correlation with UMY.

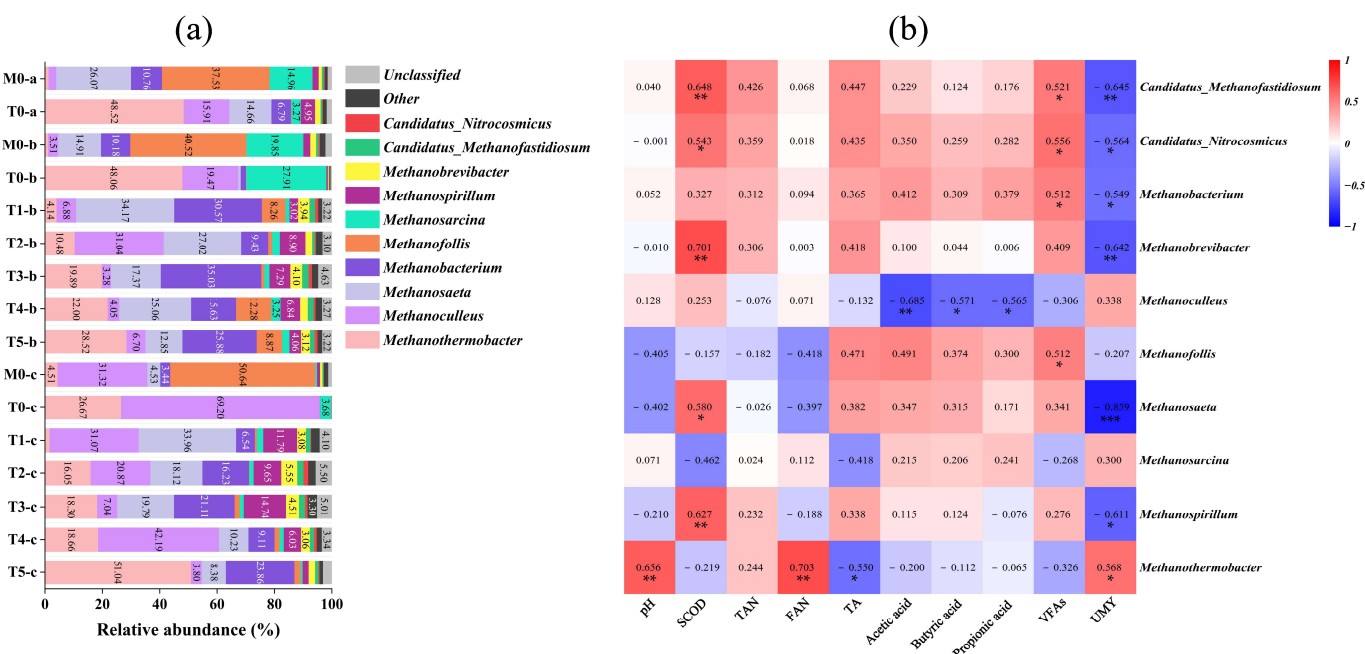

**Figure 9.** Archaeal genus structure (**a**) and correlation heatmap (**b**) of one-step abrupt temperature change from mesophilic to thermophilic. (a: on the 9th day, b: on the 27th day, c: on the 43rd day). *: $p < 0.05$, **: $p < 0.01$; *** $p < 0.001$.

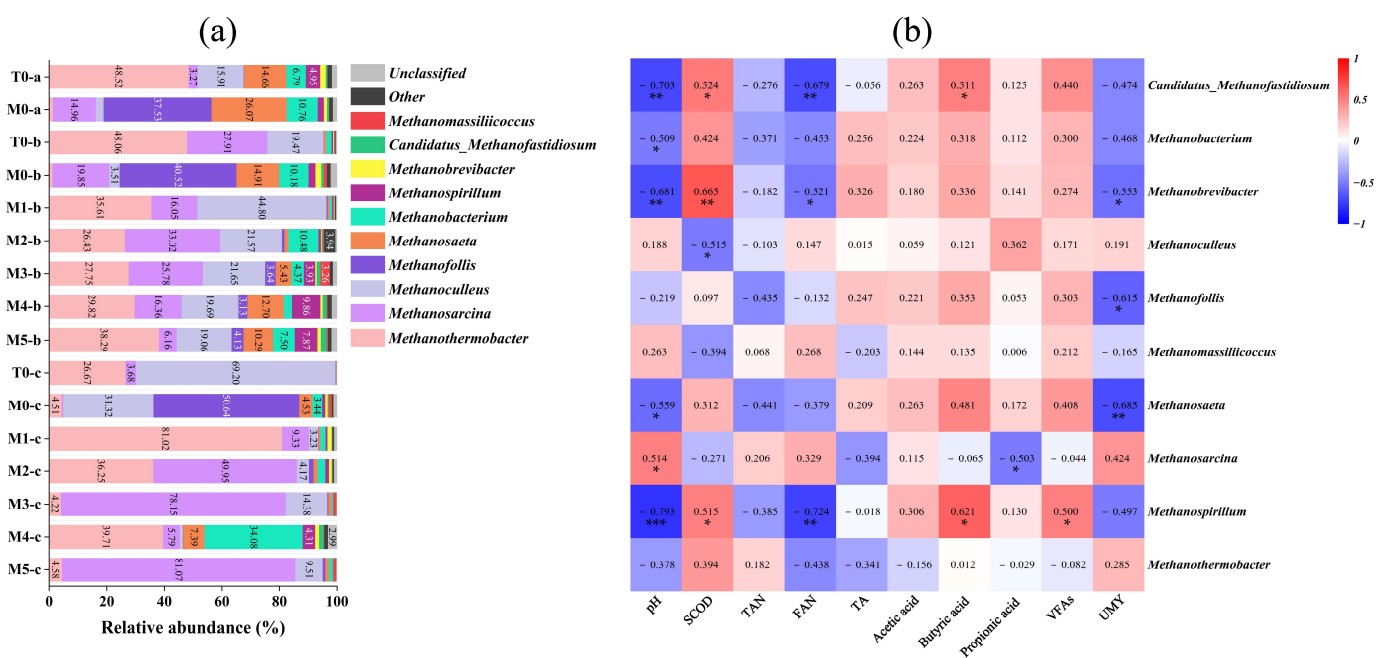

**Figure 10.** Archaeal genus structure (**a**) and correlation heatmap (**b**) of one-step abrupt temperature change from thermophilic to mesophilic. (a: on the 9th day, b: on the 27th day, c: on the 43rd day). *: $p < 0.05$, **: $p < 0.01$; *** $p < 0.001$.

## 4. Conclusions

The one-step temperature change from thermophilic to mesophilic in anaerobic digesters was more stable than that from mesophilic to thermophilic. The TAN under thermophilic conditions increased by 22.8% compared to mesophilic. Synergistic inhibition by ammonia and VFA under thermophilic was more serious than that under mesophilic

conditions. As to M-T digesters, low VFA, and high inoculum ratio could enhance digester stability after a one-step temperature change. As to T-M digesters, a low inoculum ratio (around 5%) could ensure digester stability after a one-step temperature change, which improved CMY by 9.4–433.4% compared with other inoculum rates. The results provided a basis for varying the ratio of inoculated sludge after changing anaerobic digestion temperatures for different seasons.

**Author Contributions:** Conceptualization, W.H. and Y.Z.; methodology, T.W.; validation, T.W.; formal analysis, W.H., Y.Z. and H.Z.; investigation, W.H. and Y.Z.; data curation, T.W.; writing—original draft preparation, W.H. and Y.Z.; writing—review and editing, H.Z.; supervision, T.W. and W.H.; project administration, T.W. All authors have read and agreed to the published version of the manuscript.

**Funding:** This research was funded by: National Natural Science Foundation of China [Grant No. 52360019, 51741805]. Province Natural Science Foundation of Gansu [Grant No. 22JR5RA257].

**Institutional Review Board Statement:** Not applicable.

**Informed Consent Statement:** Not applicable.

**Data Availability Statement:** Data are contained within the article.

**Acknowledgments:** We thank the following funders for their support and assistance in the experiment and publication of this article: Research and Engineering Demonstration on Integrated Technology of Municipal Sludge Low Carbon Treatment (2022-H-004), Ministry of Housing and Urban-Rural Development of the People's Republic of China (MOHURD). Study on Carbon Emission Effect of Sludge Core Treatment Process in Large and Medium-sized Urban Wastewater Treatment Plants (21230731100), Science and Technology Commission of Shanghai Municipality. Carbon Emission Reduction Technology of sludge digestion treatment in Urban Wastewater Treatment Plants (21YJKF-21) Shanghai Construction Group Co., Ltd.

**Conflicts of Interest:** Author Weijie Hu and Youfei Zhou are employed by the company "Shanghai Municipal Engineering Design and Research Institute (Group) Co., Ltd.". But for this investigation, there was no financing relationship with the company; therefore, there are no conflicts of interest. The remaining authors declare that the research was conducted in the absence of any commercial or financial relationships that could be construed as a potential conflict of interest.

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
