# Peer review of "Effects of One-Step Abrupt Temperature Change on Anaerobic Co-Digestion of Kitchen Waste with Dewatered Sludge"

_fermentation, doi:10.3390/fermentation10010005_

Round 1

Reviewer 1 Report

Comments and Suggestions for Authors

Reviewer 2 Report

Comments and Suggestions for Authors

Treatment and utilization of kitchen waste and sludge has high relevance for the practice and contributes to achieve the sustainability. Therefore, manuscript fermentation-2758247 has a relevant topic. The manuscript is generally well structured and contains interesting information not just for the science but also for the practice. The results are discussed with relevant references. The manuscript has a good quality, but the Introduction section and methodology parts need significant revision to make the manuscript more complete and clearer.

Comments, suggestions:

In the Introduction section discuss the effect of temperature, temperature change on the AD kinetics, biogas yield in more details (line 47-51).

Please define and give clearly the novelties of the study in the Introduction section.

Please give how the mixing ratio of KW and sludge selected/determined (section 2.2).

Please discuss the relevance of optimized C/N ratio and co-digestion in more details (line 44-46).

Please check the typing and stylistic errors and inconsistencies in the whole manuscript (see missing spaces in lines 26, 27, 29 etc; or phrasing in line 42-43, etc…).

Please give the C:N ration of the processed materials as well in Table 1.

Gompertz equation is used for the investigation of AD kinetics not for statistical analysis (section 2.4).

Please reconsider the title of section 3.1 (line 138). Ther are two 3.1 section title.

The visibility of Figure 1 is very poor.

The Conclusion section is too superficial and short. Please summarize the main aims, findings, and future outlooks, as well.
